# WikiWhy: Answering and Explaining Cause-and-Effect Questions

**Matthew Ho**[*] **Aditya Sharma**[*]**, Justin Chang**[*],
**Michael Saxon, Sharon Levy, Yujie Lu, William Yang Wang**
Department of Computer Science, University of California, Santa Barbara, USA
{msho, aditya_sharma, justin_chang}@ucsb.edu,
{saxon, sharonlevy, yujielu}@ucsb.edu, william@cs.ucsb.edu

## Abstract

As large language models (LLMs) grow larger and more sophisticated, assessing their "reasoning" capabilities in natural language grows more challenging. Recent question answering (QA) benchmarks that attempt to assess reasoning are often limited by a narrow scope of covered situations and subject matters. We introduce WikiWhy[1], a QA dataset built around a novel auxiliary task: explaining *why* an answer is true in natural language. WikiWhy contains over 9,000 "why" question-answer-rationale triples, grounded on Wikipedia facts across a diverse set of topics. Each rationale is a set of supporting statements connecting the question to the answer. WikiWhy serves as a benchmark for the reasoning capabilities of LLMs because it demands rigorous explicit rationales for each answer to demonstrate the acquisition of implicit commonsense knowledge, which is unlikely to be easily memorized. GPT-3 baselines achieve only 38.7% human-evaluated correctness in the end-to-end answer & explain condition, leaving significant room for future improvements.

## 1 Introduction

Error analyses of practical NLP systems in recent history demonstrate that some of the mistakes made by state-of-the-art models would be avoided by basic human intuition (Shuster et al., 2022), and some of the most challenging tasks for models are the same ones that might be trivial to human children. With modern systems' impressive performance on tasks such as grammar correction showing that manipulating language is not the issue, LLMs seem to face a fundamental lack of common sense– an understanding of everyday phenomena and how they interact with each other and the world at large. As striking gains in subjective performance on summarization, creative text generation, and apparent language understanding continue to be called into question, the development of strong benchmarks to assess reasoning capabilities for these LLMs grows more important.

One popular approach to measuring reasoning capability is through performance on question answering (QA) benchmark tasks where direct queries for information act as a straightforward examination of a system's "understanding." Classic QA datasets, however, are primarily concerned with retrieving factoids to answer questions of "Who", "What", "When", and "Where". These questions have been shown to be answerable (with high accuracy) by simple pattern-matching approaches (Wadhwa et al., 2018), thereby limiting their ability to measure the aforementioned reasoning capability. Looking to maintain the breadth of topics covered while increasing the difficulty of the QA task, researchers introduced multi-hop QA datasets like HotpotQA (Yang et al., 2018). While challenging, the task's extra complexity mostly leads to unnatural questions that can be addressed with iterated factoid retrieval and entity resolution, rather than a necessary understanding of how different entities interact. Noticeably absent in these prior datasets are "why" questions, which prompt for not factoids, but explanations– reasoning made explicit.

---

[*]Co-first authors. Author contributions listed at end of paper.
[1]https://github.com/matt-seb-ho/WikiWhy

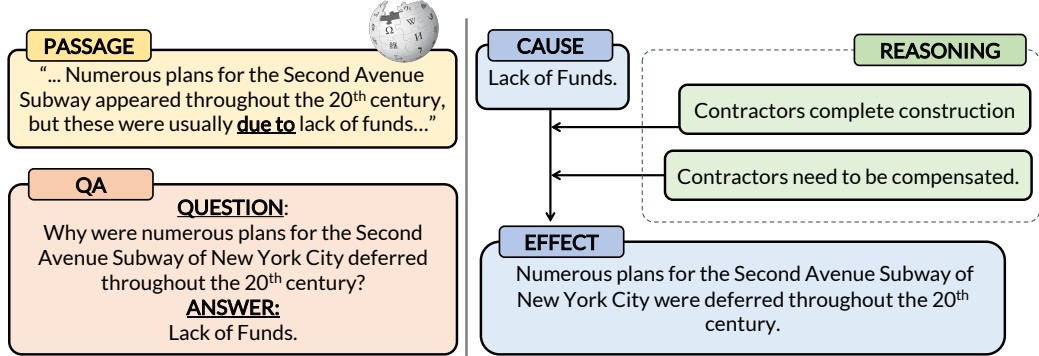

Figure 1: A simple example of an entry from WIKIWHY; a **cause** and **effect** sourced from a Wikipedia **passage**, a "why" **question** and its **answer** about this relation, and most importantly **rationale** that explains why **cause** leads to **effect**.

The task of explanation uses reasoning and produces explicit, interpretable "thought" processes. Capitalizing on these properties, this paper introduces WIKIWHY, a novel dataset containing "why" question-answer pairs. Each WIKIWHY entry contains a rationale explaining the QA pair's causal relation (Figure 1), summing to a total of 14,238 explanation elements. In the context of recent multimodal, self-supervised approaches aiming to capture intuitions unlearnable from text alone (Chadha & Jain, 2021), WIKIWHY presents an opportunity to investigate a specific kind of information absent in text: implicit commonsense assumptions. Compared to other QA datasets with rationales, WIKIWHY covers a significantly broader range of 11 topics which may prove valuable for developing the skill of applied reasoning on various specific situations.

Our experiments in explanation generation and human evaluation demonstrate that state-of-the-art generative models struggle with producing satisfying explanations for WIKIWHY cause-effect relations. Our experiments also demonstrate how our proposed task might be used to diagnose a lack of "understanding" in certain relations. Our key contributions are thus:

- We propose explanation **within** cause-effect relations as a novel problem formulation for exploring LLM reasoning ability.
- We create WIKIWHY, the first question-answering dataset focusing on reasoning **within** causal relations, spanning **11 topics**.
- We perform experiments on state-of-the-art, generative models to investigate various settings and establish baseline results with sizable room for improvement.
- We introduce idea-level evaluation metrics for free-form text (explanation) generation and a human judgment correlation analysis, demonstrating that (1) reference similarity is strongly correlated with explanation correctness, and (2) the metrics we introduced correlate with this proxy.

## 2 RELATED WORK

**Cause and Effect.** Causality has been a subject of rigorous work in various fields. In science philosophy, Pearl (2009) has contributed seminal work relating to causal models, Bayesian networks, and causal strength via interventions and counterfactuals. These ideas have even been incorporated into QA tasks through Knowledge Graph approaches, such as filtering spurious latent correlations (Sui et al., 2022). While our work emphasizes cause-and-effect, we are unconcerned with causal strength. Starting with Wikipedia-grounded relations ensures valid relations. Instead, we are interested in the information encoded into LLMs rather than augmented structures such as knowledge graphs.

**Multi-hop Question Answering.** While datasets such as HotpotQA (Yang et al., 2018) and Hy-bridQA (Chen et al., 2020) are instrumental in gauging models' ability to handle multiple sources and modalities, they are focused on iterated factoid retrieval. Although chaining multiple facts into

Table 1: A comparison of WIKIWHY with previous QA datasets relating to explanation

| Dataset | Size | Answer Type | Explanation Type | Topics | Source |
|---|---|---|---|---|---|
| CoS-E[1] | 9,500 | MCQ | 1-step | 1 | ConceptNet |
| eQASC[2] | 9,980 | MCQ | 2-step | 1 | WorldTree |
| CausalQA[3] | 24,000 | Short | None | 1 | Yahoo Finance |
| EntailmentBank[4] | 1,840 | Short | Tree | 1 | WorldTree |
| WIKIWHY | **9,406** | **Short** | **Set/Chain** | **11** | **Wikipedia** |

[1](Rajani et al., 2019), [2](Jhamtani & Clark, 2020), [3](Yang et al., 2022), [4](Dalvi et al., 2021)

a multi-hop answer is useful for products, WIKIWHY focuses on *in-filling* rationales to demonstrate reasoning.

**Visual Question Answering.** Vision and language tasks have also intersected with both QA and reasoning. The Visual Question Answering (VQA) dataset (Agrawal et al., 2015) prompts textual answers to questions about images. However, the caption-based generation leads to surface-level questions that require little reasoning ability, and the multiple-choice output format precludes explicit reasoning. The vision-based Sherlock dataset (Hessel et al., 2022) is much closer to our work, focusing on abductive reasoning (working backward from a consequence). Setting aside modality differences, WIKIWHY requires deeper reasoning with its multi-hop explanations.

**Explainable QA.** One previous approach to building explanation resources collects direct answers to "why" questions. TellMeWhy (Lal et al., 2021) features question-answer pairs tied to short story narrative contexts. The dataset skips step-wise explanations, prioritizing reading comprehension instead. On the other hand, ELI5 (Fan et al., 2019) dives deep into reasoning with long-form, detailed explanations. However, the open-endedness (compared to explaining a specific cause-effect relation) complicates evaluating candidate responses.

Another line of QA work emphasizes a rationale component as support for answer predictions. Datasets like CoS-E (Rajani et al., 2019), eQASC(Jhamtani & Clark, 2020), and EntailmentBank (Dalvi et al., 2021) focus on explanation and reasoning much like WIKIWHY, albeit with significant differences (Table 1). CoS-E's explanations for CommonsenseQA (Talmor et al., 2019) mark an important first step, but the commonsense explanations have limited depth, often requiring a single hop of reasoning. eQASC and EntailmentBank feature richer explanations with more complex structure, tightly focusing on grade school level science facts. Regarding structure, fixed-length rationale in CoS-E, eQASC, FEVER (Thorne et al., 2018), and e-SNLI (Camburu et al., 2018) capture less granularity, while entailment trees accept limitations in scale and naturalness in exchange for complete ordering information. Previous datasets tend towards retrieval tasks with eQASC's corpus of all rationale sentences and EntailmentBank's collection of root causes. Retrieval enables simple evaluation, at the cost of decreased difficulty, the possibility for exploiting spurious artifacts, and reduced debugging opportunity.

## 3 BACKGROUND

### 3.1 WHY FOCUS ON "WHY" QUESTIONS?

"Why" questions are underrepresented in other QA datasets. Users tend to ask straightforward questions that use words like "who", "what", "when" or "where." Questions of this more common form have simple answers that state standalone facts which may be elaborated but do not require explanation. Consider the pair, *"Q: What is the fifth most abundant metal in the Earth's crust? A: Calcium."* The answer is straightforward.

In contrast, a "why" QA-pair encodes a cause-effect relation. Take, for example, *"Q: Why can't I eat calcium metal? A: Calcium reacts exothermically with water and acids*. This pair encodes the causal relation *"calcium has an exothermic reaction with water, therefore eating calcium is not advised."* (Figure 2). The answer to a "why"-question is an explanation itself (the reaction being exothermic explains the toxicity), but we can take it a step further and ask "why" *again* to request the understanding, or intuition, of this process. While there are some processes at the edge of human

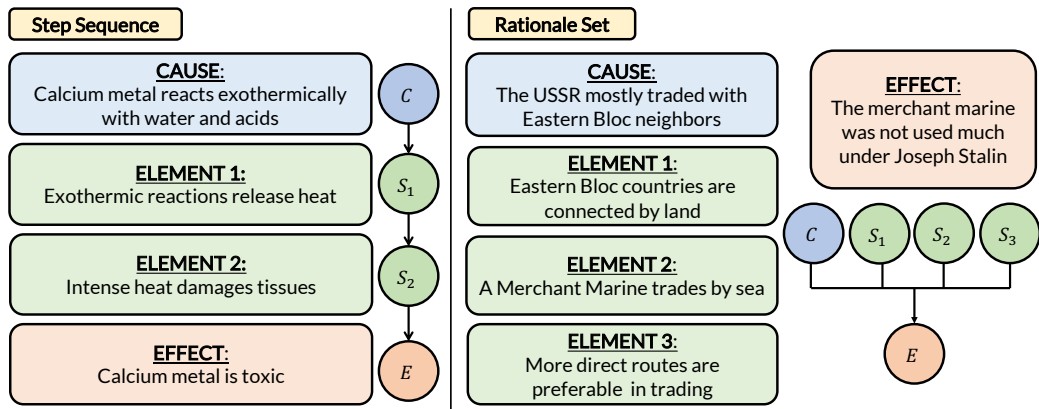

Figure 2: Explanation topologies in WIKIWHY mainly vary between a sequence of intermediate conclusions (chain-like) and a set of rationale that combine with the original cause to entail the final effect.

understanding or taken as axioms, we assert that there are valid explanations for most processes due to the layered nature of human understanding. This extra step is especially worth taking since it allows WIKIWHY to not only test if a model "knows" that *"an exothermic water reaction makes calcium toxic"* but also if it "understands" the underlying mechanics of why that is the case.

## 3.2 TASK FORMULATION

Formally defined in §5, we propose a **generative** explanation task. Previous works have made strides in assessing reasoning through multiple choice (Lu et al., 2022), retrieval (Asai et al., 2019), and partial generation (Dalvi et al., 2021). While these works are undoubtedly crucial towards the end goal of understanding and reasoning, their task formulations have some drawbacks. Referring back to education, studies on human students have shown that multiple choice questions "obscure nuance in student thinking" (Hubbard et al., 2017). Likewise, a selection decision can be correct for retriever systems but for the wrong reasons. Augmenting multi-hop factoid questions with an additional task of selecting the relevant supporting facts from the context passage, Inoue et al. (2020) emphasizes that interpretability is lost in the absence of explanation. Furthermore, text generation to combine existing ideas is arguably a different task than generating from scratch. The field of psychology defines recall (mental retrieval of information) as a distinct process from recognition (mental familiarity with the cue) (Mohr et al., 1989). Neural nets' biological inspiration suggests that there might be a similar difference between cue-aided retrieval and freeform generation. In the context of NLP, we are interested in the implicit understandings and assumptions embedded in LLMs and hypothesize that an entirely generative approach is most conducive to this study.

## 3.3 EXPLANATION STRUCTURE

Explanations come in various structures, as seen in the typology defined by Ribeiro et al. (2022). Shown in Figure 2, our work focuses on a subset of said typology. WIKIWHY includes two structures that explain cause-and-effect relations: (1) multi-hop step sequences and (2) rationale sets. While the chain structure adds intermediate conclusions between cause and effect, rationale sets contain elements that support the relation from without. The rationale set topology acts as our general, catch-all case that other structures can be condensed to. Since our data collection procedure promotes a stepwise, ordered approach, we also consider the sequential topology to respect the structure exhibited in applicable explanations. We forego the unstructured approach as even limited structure helps bring freeform generated text evaluation within reach. Finally, we opt against pursuing the most complex entailment tree organization to maintain naturalness and facilitate crowdsourcing scalability.

## 4 DATASET

### 4.1 DATA COLLECTION

The objective of WIKIWHY is to present a high-quality, challenging dataset of QA pairs with corresponding causes, effects, and explanations. We developed an extensive data collection and validation pipeline around Amazon Mechanical Turk, depicted in Figure 5 (appendix). For each stage involving crowdsourced annotations, we perform rigorous worker-level quality control to ensure the dataset's quality. The exact procedures are detailed in subsection A.2 in the appendix.

**Preprocessing.** We begin with English Wikipedia's corpus of "Good Articles,"[2] whose strict criteria of verifiability and neutrality (among others) ensure that WIKIWHY does not evaluate models on misinformation or opinionated views. From these articles, we extract passages containing causal relations using causal connectives. We selected a list of causal keywords (Appendix, §subsection A.1) from a more extensive set of causal connectives as their presence in a passage guarantees the existence of a cause and effect relation—some excluded connectives such as "since" or "as" are highly prevalent but are not necessarily causal. The presence of a causal word pattern on its own is a very simple heuristic—in the subsequent collection steps, we hired crowdworkers to ensure the quality of each sample.

**QA Synthesis (Stage 1).** Randomly sampled preprocessed Wikipedia passages containing potential causal statements were shown to qualified Amazon Mechanical Turk (MTurk) workers (see ethics statement for details), who were tasked with extracting the highlighted causal relation from the passage and re-framing it as a "why" question when possible. While automatic cause-effect relation extraction has seen recent progress (Yang et al., 2022), this human intelligence task (HIT) remains vital for two reasons. First, we find that quality in cause-effect is crucial for meaningful and valid explanations in the following stage. More importantly, we depend on human annotators to add sufficient context to the text of the cause, effect, and question to disambiguate them. This enables the question and cause-effect relation to be presented to models without the context we prepared (e.g., "Why was the river diverted?" is unanswerable without additional context). This feature is key to enabling WIKIWHY to assess the information and ideas within LLMs as opposed to whatever may be present in the context.

**Explanation Synthesis (Stage 2).** After verifying the quality of the examples, we prompt crowd workers to explain cause-effect pairs from stage 1. To encourage structured explanation, we supply an interface that allows sentences or ideas to be entered one at a time in separate fields. Though the input pairs should be context-independent, we provide the original passage as an aid for understanding the topic. Furthermore, we provide the link to the source article to encourage explanations leveraging topic-specific information in addition to commonsense knowledge.

### 4.2 DATASET DESCRIPTION

**Entry Contents.** In addition to the main fields of the question, answer, and explanation, each dataset entry contains the underlying relation's cause and effect, the passage the question was extracted from, the article the passage is from, and Wikipedia's topic categorization for that article.

**Topic Diversity.** WIKIWHY improves upon other datasets due to its ability to examine reasoning proficiency across a broader range of concepts (Table 9 in Appendix contains examples from the six most frequent topics).

**Rationale.** The statistics for the reasoning component are shown in Appendix Table 11. On average, each rationale contains **1.5137** elements. Figure 4 (Appendix) shows a histogram of rationale length by sentence count. WIKIWHY includes a range of rationale lengths, with more than one-third of examples (36%) containing two or more reasoning steps.

---

[2]https://en.wikipedia.org/wiki/Wikipedia:Good_articles/all

## 5 EXPERIMENTS

### 5.1 EXPERIMENTAL SETTINGS AND MODELS

**Task Notation**   Let $C$ be a cause clause; $E$ be an effect clause corresponding to $C$; $Q$ be a question equivalent to "Why is it the case that $E$?"; $A$ be the answer to $Q$ [3]; $X$ be the explanation = $(S_1, S_2, \ldots, S_k)$ where $S_i$ is a sentence such that:

$$C \wedge S_1 \wedge S_2 \wedge \ldots \wedge S_k \vdash E$$

**Task 1: Question Answering (QA). Input = $Q$, Output = $A$.**   For thoroughness, we confirm high performance on Task 1 (Standard QA) in the open-book setting. For this set of experiments, we use the classic approach of breaking the task into separate retrieval and reading comprehension phases. We experiment with BM25 (Robertson et al., 2009) and Dense Passage Retriever (DPR) (Karpukhin et al., 2020) as our document retriever, using their Pyserini implementations (Lin et al., 2021). Using the Natural Questions (Kwiatkowski et al., 2019) encoder, as in the original DPR paper, we build custom indices around segments from the subset of Wikipedia Articles shown to workers at collection time. For reading comprehension, we experimented with RoBERTa (Liu et al., 2019) and Big Bird (Zaheer et al., 2020) QA models. We also fine-tune a Fusion-in-Decoder (FiD) (Izacard & Grave, 2020) model (80-10-10 split; default configurations), hypothesizing the decode-time combination of ideas could better model cause-effect relations.

The performance was unsurprisingly high, with BM25 achieving a high Top-1 Accuracy score of 0.810 in retrieval and FiD reaching a mean BERT-f1 of 0.78 (Table 7 in Appendix). While retrieving the appropriate Wikipedia passage relating to some topic is straightforward, we found that producing an explanation of comparable quality to our gold rationales was difficult for the models we tested.

**Task 2: Explanation Only (EO). Input = $(C, E)$, Output = $X$.**   First, we examine task 2: generating an explanation given an initial cause-effect pair. Given their stronger zero-shot generalization (Wang et al., 2022), we choose decoder-only models for our baselines. In this vein, we investigate the few-shot abilities of GPT-3 (Brown et al., 2020) with OpenAI's most capable model, DaVinci-002, at otherwise default settings. To better coax out the intermediates between cause and effect, we conduct prompt engineering over Wei et al. (2022)'s Chain of Thought method. Our exemplars are shown in Figure 6.

We also make use of WIKIWHY's scale for fine-tuning GPT-2 (Radford et al., 2019). In this set of experiments, we attempt to balance improving GPT-2's understanding of the task's structure while preserving the model's "intuitions" for examination. We train GPT-2 for ten epochs using the training split ($\approx 80\%$ of the data) and Adam (Kingma & Ba, 2014) optimizer with standard hyperparameters (learning rate: $\gamma = 0.001, \beta_1 = 0.9, \beta_2 = 0.999, \epsilon = $ 1e-8, decay: $\lambda = 0$). For this tuned model we introduce special delimiter tokens `<cause>`, `<effect>`, and `<explanation>` in addition to the beginning and end tokens `<bos>` and `<eos>`. To support the delimiters and help the model distinguish the segments, we add token type embeddings (marking cause, effect, and explanation) as part of the preprocessing phase. At decoding time, we experiment with multiple temperatures.

**Task 3: Answer and Explanation (A&E). Input = $Q$, Output = $(A, X)$.**   To investigate the performance of jointly predicting an answer and explanation given only a "why" question, we carry forward with our best performing baseline from the **EO** task, chain-of-thought prompted GPT-3. The first setting in this experiment set tasks a single model with the full end-to-end procedure. Once again, we utilize Chain-of-Thought prompting, albeit with a modified prompt that also requests an answer to handle the different input format. Considering the impressive performance of existing IR techniques on the **QA** task described above, we also study an additional setting incorporating the **QA** task. In the "pipeline" setting, the explainer model still lacks access to the ideal answer (the explanation's starting point) but benefits from a reader model's access to the original context. Here we combine our strongest performing approaches to the **QA** and **EO** tasks to make a 3-step pipeline of retrieval (BM25), reading (FiD), and explanation (GPT-3).

### 5.2 AUTOMATIC EVALUATION METRICS

While the still developing area of text generation has measures and proxies for similarity that help with simple sequences, comparing reasoning sequences or rationale sets requires more involved

---

[3]Note that $Q$ is a query that provides $E$ and is correctly answered by $C$, $C = A$.

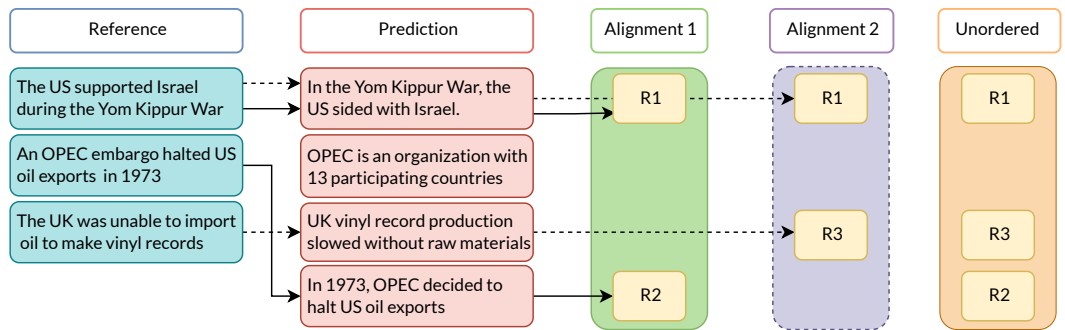

Figure 3: Alignment example for sentence-level metrics. Ordered evaluation uses the longest common subsequence as shown by alignment 1 and 2. The final alignment's length is used to compute F-score metrics.

measures. With the two topologies introduced in §3.3 in mind, we propose two related metrics, unordered and ordered, to handle sets and sequences, respectively.

**Unordered Evaluation.** This first approach compares the ideas contained in the predictions and references. First, we split predicted and reference explanations into "ideas" or "steps" by sentence. We then compute a matrix of pairwise similarity scores before using a threshold to classify "matches". Since a single prediction sentence may contain multiple reference ideas, we keep separate counts of precise prediction steps and covered reference steps. These counts are then micro-averaged for the test set's overall precision, recall, and F1 scores.

**Ordered Evaluation.** To respect the structure of multi-hop explanations, we penalize incorrectly ordered explanations. Here, we use the previously generated pairwise score matrix and its alignments to generate all possible assignments of prediction sequence elements to reference elements. As demonstrated in Figure 3, we compute the length of the longest common subsequence (LCS) between a prediction alignment against the reference labels for each candidate assignment. This length becomes the count of correctly incorporated structural elements– true positives. Note that the LCS alignment discounts repeated ideas in the prediction.

**Metric Validity.** To understand the usefulness of our constructed metrics, we compare them against human judgements. A panel of 3 undergraduate students compared pairs of predictions and references on two binary scales: (**1. Similarity**) "Is the prediction similar to the reference?" and (**2. Correctness**) "Is the prediction a valid or correct explanation of the cause-effect pair?" Summing the panelist scores for each pair, we found a strong correlation ($r = 0.82$) between the similarity and correctness judgement. This validates comparison with WIKIWHY gold explanations as a useful proxy for explanation quality. Our proposed sentence-level processing incorporates the intuitions of checking for completeness with recall and penalizing over-explanation with precision.

Further, we use a single-explanation version of F-score to compare this proposed automatic metric with human judgement (the proposed F-score measures aggregate through the whole dataset). With this variation, we find a modest correlation ($r = 0.35$) between ordered F1 and similarity, among other weaker correlations.

Besides supporting our proposed methods, this correlation analysis also enabled a data-driven approach to calibrating our similarity metric and match criteria. For each similarity metric, we selected a starting point through manual inspection of prediction-reference-similarity triples (which threshold value divides "genuine" from mistaken similarity) and used correlation for refinement. After trials with BLEURT (Sellam et al., 2020) and BERTScore (Zhang et al., 2020), different underlying models and different match thresholds, we selected BERTScore using a large DeBERTa (He et al., 2021) model (`microsoft/deberta-xlarge-mnli`) at a threshold of 0.64.

### 5.3 HUMAN EVALUATION

Recent studies by Goyal et al. (2022) show that automatic metrics may not reliably evaluate results produced by models with few-shot capabilities like GPT-3. In light of this, we supplement our auto-

matic evaluation with an additional human evaluation. We first evaluate each setting in each experiment using the binary correctness scale (see criteria definition below). Following this evaluation, we select the highest scoring explanations for each set of experiments for additional fine-grained evaluation. For each human evaluation task, we present a panel of three undergraduate students a random sample of 50 entries from each setting and the following binary True/False criteria guidelines:

- **Correctness**: Mark true if and only if the explanation is both complete and satisfying.

- **Concision**: Mark true if the explanation says everything it needs to say and nothing more. Mark false if extra information is included.

- **Fluency**: Is the explanation writing fluent? Mark false if there are any mechanical mistakes.

- **Validity**: Does the explanation make logical sense? Ignore whether or not the explanation successfully explains the cause/effect relation. Mark false if the explanation contains any illogical or untrue conclusions.

- **Win/Tie/Lose**: Compare the generated explanation against the provided reference (WIKIWHY gold explanation). Mark Win if you prefer the generated explanation, Tie if you have no preference, and Lose if you prefer the reference explanation.

## 5.4 RESULTS

**Fine-Grained Human Evaluation.**  With our human evaluation experiments, we find significant room for improvement across the board. Our strongest baseline, GPT-3 with greedy decoding, produced explanations judged to be satisfactory only 66% of the time in the most favourable setting of Task 2: EO (Table 3). Moreover, these explanations were judged to be worse than the gold reference 58% of the time. These results from our strongest baseline leave plenty of room to improve upon and motivate future work on this reasoning task.

**Decoding.**  Our experiments show increased performance with lower temperature sampling and best results with greedy decoding (Table 2). This aligns with existing notions of higher temperatures better suiting "creative," open-ended tasks as opposed to more grounded ones. Explaining, as we hypothesize, relies more on the embedded assumptions in a generative model rather than the tenuous associations made more likely at higher temperatures.

**Model Differences.**  We find that GPT-3 significantly outperforms GPT-2. Comparing GPT-3's output against its predecessor's strongest setting shows increases in both ordered and unordered F1 scores by over 50%. Despite benefiting from fine-tuning and additional structural support from token type embeddings, GPT-2's explanations are lacking compared to GPT-3's few-shot explanations using only 4 exemplars. We find that GPT-2's statements are often not only incomplete/unsatisfying for explaining the cause-effect relation at hand but also simply invalid. 94% of GPT-2's statements were deemed worse than WIKIWHY's gold references. The only area GPT-2 outperformed GPT-3 was in concision, however this is more a demerit of GPT-3 rather than a merit of GPT-2. We found that GPT-3 tended to occasionally add unnecessary detail to its explanations, often needlessly defining one of the entities in the prompt.

**Answer & Explanation.**  On the A&E task, we find results that align cleanly with preconceived intuitions. Our baseline model is able to better handle explanations from points A to B when A is fixed and provided. Requiring the same procedures to generate more output creates more variance as incorrect or alternative starting points mislead the remaining generation. The "pipeline" setting strengthens this trend, as the better-informed answer generation allows for a higher quality explanation. This setting, simulating a three-step process with different models handling each step, demonstrates an intermediate performance between having the oracle-provided answer and requiring the explainer to manage the entire process. The "validity" criteria of our human evaluation is especially interesting under this setting, where the model's input excludes the correct answer (the cause). While the majority of the end-to-end setting's explanations were marked incorrect or unsatisfying, a notable proportion was still marked as having a valid chain of reasoning. This supports the chain-of-thought premise that short, logical strides are more manageable for LLMs, but the approach is still insufficient for generating satisfactory explanations of target phenomena.

**Explanation Failure.**  A typical error observed in GPT-3's predictions is repeating the cause-effect relation. To explain why `[A]` leads to `[B]`, GPT-3 might only write "`[B]` because `[A]`" or another

Table 2: Baseline Performance on Explanation Tasks (EO = Explanation-Only, A&E: Answer and Explanation). For Task 3, the Single Model setting has the generative model complete the end-to-end task in a single pass. The Pipeline setting allows each stage to be handled separately (QA is handled by BM25+FiD and explanation is done by GPT-3). Human evaluation was done with on a binary scale (correct/incorrect) and we report the proportion of correct evaluations.

| Experiments | Unordered | | | Ordered | | | Human |
|---|---|---|---|---|---|---|---|
| | Precision | Recall | BERT-f1 | Precision | Recall | BERT-f1 | Correct |
| **Task 2: EO** | | | | | | | |
| GPT-2 | | | | | | | |
| *Greedy* | 0.249 | 0.196 | 0.220 | 0.239 | 0.179 | 0.204 | **0.100** |
| $T = 0.50$ | 0.218 | 0.164 | 0.188 | 0.194 | 0.146 | 0.166 | 0.065 |
| $T = 1.00$ | 0.072 | 0.056 | 0.063 | 0.071 | 0.054 | 0.062 | 0.064 |
| GPT-3 | | | | | | | |
| *Greedy* | 0.347 | 0.388 | **0.366** | 0.307 | 0.355 | **0.329** | **0.660** |
| $T = 1.00$ | 0.326 | 0.356 | 0.340 | 0.291 | 0.328 | 0.308 | 0.481 |
| **Task 3: A&E** | | | | | | | |
| GPT-3 | | | | | | | |
| *Single-Model* | 0.092 | 0.095 | 0.094 | 0.082 | 0.092 | 0.087 | 0.140 |
| *Pipeline* | 0.229 | 0.233 | **0.231** | 0.211 | 0.220 | **0.215** | **0.387** |

Table 3: Human evaluation. Overall correctness is marked on a binary scale– an explanation is complete and satisfying or not. Concision penalizes for repeated or unnecessary information, fluency evaluates grammar, and validity measures if the generated sequence makes logical sense regardless if it correctly explains the relation. For Win/Lose/Tie, annotators compared the generations against WIKIWHY's gold references.

| Setting | Fine Grained Human Evaluation | | | | | | |
|---|---|---|---|---|---|---|---|
| | Correctness | Concision | Fluency | Validity | Win (↑) | Tie | Lose (↓) |
| **GPT-2: EO** | 0.100 | 0.880 | 0.860 | 0.520 | 0.040 | 0.040 | 0.920 |
| **GPT-3: EO** | **0.660** | 0.680 | 1.00 | 0.960 | 0.080 | 0.360 | 0.580 |
| **GPT-3: A&E** | 0.140 | 0.680 | 0.900 | 0.720 | 0.080 | 0.100 | 0.820 |

semantically equivalent formulation. This pattern may be explainable with a fine-tuned baseline where annotation errors of the same kind might have slipped into the training set, but GPT-3 was prompted with hand-picked exemplars with no such mistakes. Furthermore, we observe successful explanations on some inputs we expect to be more difficult alongside errors on relatively less challenging inputs. These observations, together with the consistently high fluency scores showing syntactic competence, seem to indicate a reasoning failure as opposed to a systematic "misunderstanding" of the task at hand. Per the original goal of better understanding what and how LLMs "understand" the world, this might indicate a gap in commonsense: that GPT simply memorized the fact that [A] leads to [B].

## 6 CONCLUSION

With this paper, we release WIKIWHY, a Question-Answering dataset enabling the analysis and improvement of LLMs' reasoning capability. We propose explanation **between** grounded cause-effect pairs to distinguish memorization of the relation from a genuine understanding of the underlying mechanics. Compared to related works on explainable QA, our explanation format finds a natural middle ground that balances **complexity** and **depth**, allowing our crowdsourcing methods to produce thought-provoking examples while being highly scalable. We exploit this scalability to cover topics previously overlooked by other explanation datasets and demonstrate our proposed task to be difficult with strong baselines (our experiments feature models failing to produce satisfying explanations even under ideal conditions). Finally, we motivate the development of new automatic metrics that are better able to handle the complexities of generated reasoning.

## ETHICS STATEMENT

For data collection, our listing required workers to have a high HIT approval rating ($\geq 96\%$) and be located in English speaking regions (Australia, Canada, New Zealand, the United Kingdom, and the United States). The average hourly pay is 12.00 dollars, which exceeds the income requirements proposed in the human subjects research protocols. The project is classified as exempt status for IRB. Our interfaces include notices that we are collecting information for dataset creation, consent forms, and a link for inquiries and concerns. Our MTurk interfaces are displayed in the §A. Due to the experimental nature, limited production applicability, and relatively small dataset scale, we believe the potential for misuse or harm is negligible.

## REPRODUCIBILITY STATEMENT

We publically release our dataset and codebase at `https://github.com/matt-seb-ho/WikiWhy` containing the model tuning procedures, settings, few-shot prompts, and evaluation script.

## AUTHOR CONTRIBUTIONS

Matthew Ho developed the crowdsourcing interfaces, performed data cleaning, ran baseline retrieval/reading/explanation experiments, designed and implemented evaluation procedures, and drafted the paper. Aditya Sharma contributed to crowdsourcing interfaces, performed data cleaning, organized dataset statistics, and created tables and designed figures. Justin Chang developed the data collection infrastructure, the validation web app, and ran OpenAI baselines. Michael Saxon, Sharon Levy, Yujie Lu, and William Wang advised, guided, and ideated. All authors edited the manuscript.

## ACKNOWLEDGEMENTS

Thank you to Early Research Scholars Program (ERSP) advisors Diba Mirza and Chinmay Sonar, and Wenhu Chen for further advice. Thank you to Alex Mei for comments on the manuscript, and Ed and Lauren for input on data collection. We thank the reviewers for helpful feedback. This work is supported by the Amazon AWS AI/ML Research Award and AWS Cloud Credit for Research. MH, AS, and JC were supported by ERSP under National Science Foundation (NSF) Grant 1821415 as well as generous donations from an anonymous source towards undergraduate research at UCSB. AS was supported by College of Creative Studies Traveling Undergraduate Research Fellowship (TURF). MS was supported in part by the NSF Graduate Research Fellowship, NSF Grant 1650114. This work was also supported by the NSF Award 2048122. The views and conclusions contained in this document are those of the authors and should not be interpreted as representing the official policies, either expressed or implied, of the U.S. Government. The U.S. Government is authorized to reproduce and distribute reprints for Government purposes notwithstanding any copyright notation herein.

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

## A  APPENDIX

### A.1  DATA COLLECTION

Our corpus consists of the entirety of the English Wikipedia, snapshotted on 23 May 2022. Wikipedia presents a list of curated "Good Articles", which have been nominated and reviewed to fit the "Good Article Criteria". Articles from this category are guaranteed to have correct spelling and grammar, as well as clear and concise diction. Our final keyword list includes: "because", "due to", "therefore", "consequently", "resulted in", "resulting in", and "as a result".

## A.2 DATA COLLECTION VALIDATON

Each stage in our data collection process is followed by two additional validation layers. For Stage 1, workers are prohibited from submitting more than 20 entries until their annotations have been manually validated. The annotation result passes through another phase of manual validation to ensure that the quality is kept up after workers' initial submissions are accepted by quality control. For Stage 2, we track a separate list of qualified workers for explanation quality.

Similar to Stage 1, Stage 2's initial submit limit (the "speed bump") is 10. Undergraduate students manually reviewed the examples from stage-2-qualified workers. These panelists were instructed and shown demonstrations of marking explanations as satisfying or not and correcting minor errors for slight quality improvements. While manually approved workers write each WIKIWHY explanation, these hand-reviewed samples ultimately comprise the test and development sets. The continuous flow between stages is enabled by a backend system we implemented to maintain a database of submissions. This system serves inputs to both MTurk interfaces, as well as the front-end validation interfaces provided to the undergraduate panelists.

## A.3 ADDITIONAL RESULTS

We include additional evaluations of our generated explanations using simple metrics. Table 4 shows performance on the **EO** task, and Table 5 show performance on the **A&E** task. We also include results from the QA task in Table 7 and Table 8. Automatic evaluation on individual topics categories are included in Table 10.

## A.4 CROWD WORKER INTERFACE

Figure 7 and Figure 8 display the interfaces for the first and second stages respectively. In addition to the list of requirements, we provide examples and tips to further clarify our expectations. The passage is displayed with a link to the full article so workers can view the complete context if needed. Every passage contains a highlighted causal connective, allowing workers to quickly scan and skip irrelevant portions. Each passage is retrieved from our custom database through our API. If the passage is too difficult for the worker to understand or lacks a cause-effect relation, the worker can click the button below for another random passage.

Table 4: Explanation Evaluation Results of WIKIWHY dataset according to the following metrics: SacreBLEU (S-BLEU) Post (2018), Word-Mover's distance (WMD) Sato et al. (2021), Sentence Mover's Similarity Metrics (SMS) Clark et al. (2019), BERT-f1 Score Zhang et al. (2020), ROUGE-1, ROUGE-2, and ROUGE-L (all ROUGE-f1 Scores Lin (2004) averaged). SMS is scaled by 1000 for readability.

| Model | Fine-tuned GPT-2 vs. Few-shot GPT-3 | | | | | | |
|---|---|---|---|---|---|---|---|
| | S-BLEU | WMD | SMS | BERT-f1 | ROUGE-1 | ROUGE-2 | ROUGE-L |
| **GPT-2** | | | | | | | |
| *Greedy* | 0.042 | 0.541 | **15.81** | 0.773 | 0.212 | **0.057** | 0.184 |
| *Temp 0.5* | 0.037 | 0.540 | 15.30 | 0.770 | 0.198 | 0.047 | 0.169 |
| *Temp 1.0* | 0.022 | 0.536 | 13.25 | 0.760 | 0.161 | 0.022 | 0.134 |
| **GPT-3** | | | | | | | |
| *Temp 1.0* | **0.055** | **0.555** | 14.93 | **0.792** | **0.240** | **0.057** | **0.199** |

Table 5: **GPT-3** explanation results with various input settings: Ideal- gold cause/answer, Well-Selected- provided cause/answer predicted by best-performing reader model (**FiD**), End-to-end-provided only question/effect (**GPT-3** completes end-to-end task)

| Model | GPT-3 Prompt Input Experiments | | | | | | |
|---|---|---|---|---|---|---|---|
| | S-BLEU | WMD | SMS | BERT-f1 | ROUGE-1 | ROUGE-2 | ROUGE-L |
| **Input Setting** | | | | | | | |
| *Ideal* | **0.055** | **0.555** | **14.93** | **0.792** | **0.240** | **0.057** | **0.199** |
| *Well-Selected* | 0.030 | 0.546 | 13.27 | 0.776 | 0.203 | 0.049 | 0.149 |
| *End-to-end* | 0.023 | 0.542 | 13.22 | 0.768 | 0.200 | 0.038 | 0.144 |

Table 6: WIKIWHY dataset contains a diverse set of 11 genres. The raw counts of topic themes in articles is presented in the second column. The relative frequency is the percentage of articles in CausalQA sub-sampled from the *Good* Wikipedia articles list.

| GENRES | RAW # | FREQ. |
|---|---|---|
| AGRICULTURE | 131 | 0.436 |
| ARTS | 577 | 0.396 |
| ENGINEERING | 952 | 0.336 |
| GEOGRAPHY | 754 | 0.624 |
| HISTORY | 1023 | 0.433 |
| LITERATURE | 455 | 0.340 |
| MATHEMATICS | 27 | 0.227 |
| MEDIA | 1773 | 0.399 |
| MUSIC | 1070 | 0.229 |
| NATURAL SCIENCES | 2952 | 0.768 |
| PHILOSOPHY | 302 | 0.465 |

Table 7: Document Retrieval for WIKIWHY. **BM25** consistently outperforms **DPR**.

| MODEL | WIKIWHY | |
|---|---|---|
| | Top-1 Acc | MRR |
| **BM25** | **0.810** | **0.858** |
| **DPR** | 0.340 | 0.448 |

Table 8: Answer Evaluation Results for WIKIWHY dataset. Stage 1: **RoBERTa**, **BigBird**, and **FiD**. **FiD Gold** is fine-tuned on 80% train split & evaluated on 10% dev split.

| MODEL | WIKIWHY | | |
|---|---|---|---|
| | S-BLEU | BERT-f1 | WMD |
| **RoBERTa** | | | |
| *Gold* | 0.246 | 0.860 | 0.637 |
| *BM25* | 0.214 | 0.832 | 0.620 |
| **BigBird** | | | |
| *Gold* | 0.258 | 0.825 | 0.615 |
| *BM25* | 0.223 | 0.802 | 0.602 |
| **FiD** | | | |
| *Gold* | 0.373 | 0.863 | 0.658 |
| *BM25* | 0.259 | 0.827 | 0.617 |

Table 9: Examples from 6 most frequent topics covered in WIKIWHY. $c$ denotes cause, $e$ effect, and $s_i$ the $i$th rationale sentence.

| Genres | | Example |
|---|---|---|
| Geography | $c$ | The geographic isolation of the Hupa homeland |
| | $s_1$ | The Hupa's homeland was separated by bodies of water or mountains |
| | $s_2$ | Not many people could get to the Hupa's homeland |
| | $e$ | The Hupa had few interactions with early European explorers up to the 19th century |
| Literature | $c$ | Increased language contact in the globalizing world |
| | $s_1$ | Increased contact between people requires increased communication |
| | $s_2$ | Speaker of uncommon languages switch to more common languages |
| | $s_3$ | Switching away from uncommon languages leads to them being forgotten |
| | $e$ | Many small languages are becoming endangered as their speakers shift to other languages |
| Media | $c$ | Seeing the Castle of Cagliostro entrenched in Yamazaki that Japan can make high-quality films |
| | $s_1$ | Viewing The Castle of Cagliostro inspired Takashi Yamazaki |
| | $s_2$ | Out of national pride, Takashi Yamazaki followed a model that he believed would produce quality films |
| | $e$ | Director Takashi Yamazaki modeled his 2019 film Lupin III: The First after The Castle of Cagliostro |
| Music | $c$ | The duration of Hotel California was longer than songs generally played by radio stations |
| | $s_1$ | Most songs are only 3-4 minutes long |
| | $s_2$ | Hotel California is over 6 minutes |
| | $s_3$ | People would not want to listen to same song on radio for that long |
| | $e$ | Don Felder had doubts about the 1997 Eagles song Hotel California |
| Natural Sciences | $c$ | The thermal stress at dawn and dusk |
| | $s_1$ | The thermal temperatures change so drastically the rocks expand and contract |
| | $s_2$ | This process weakens the structural integrity of the rocks |
| | $e$ | The boulders on Ceres are brittle and degrade rapidly |
| Technology | $c$ | The use of coal power in Turkey |
| | $s_1$ | Burning coal leads to air pollution |
| | $s_2$ | Air pollution causes sickness and early death |
| | $s_3$ | Sick and dead people cannot work |
| | $e$ | 1.4 million working days were lost across the population of Turkey in 2019 |

Table 10: Explanation performance (unordered f1) over the most frequent topics. We GPT-2 under the greedy setting and GPT-3 under the same defaults as Table 2

| | **Most Frequent Genres** | | | | | | |
|---|---|---|---|---|---|---|---|
| | ARTS | GEOG | HISTORY | MEDIA | MUSIC | SCIENCE | TECH |
| **Models** | | | | | | | |
| GPT-2 | 0.256 | 0.221 | 0.202 | 0.161 | 0.239 | 0.252 | 0.236 |
| GPT-3 | 0.412 | 0.372 | 0.341 | 0.335 | 0.301 | 0.371 | 0.333 |

Table 11: WikiWhy Summary Statistics

| **WikiWhy Statistics** | |
|---|---|
| # of Train | 7,397 |
| # of Dev | 1,004 |
| # of Test | 1,005 |
| # of Rationale | 9,406 |
| # of Rationale Elements | 14,238 |
| Avg. # Rationale Length | 1.5137 |
| Avg. # Tokens per Element | 16.697 |

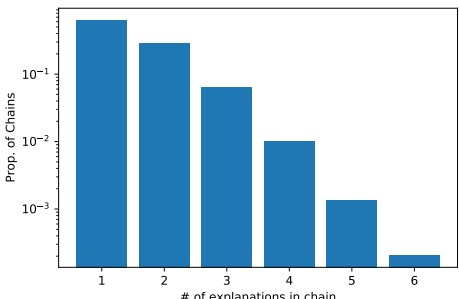

Figure 4: Rationale Length Distribution

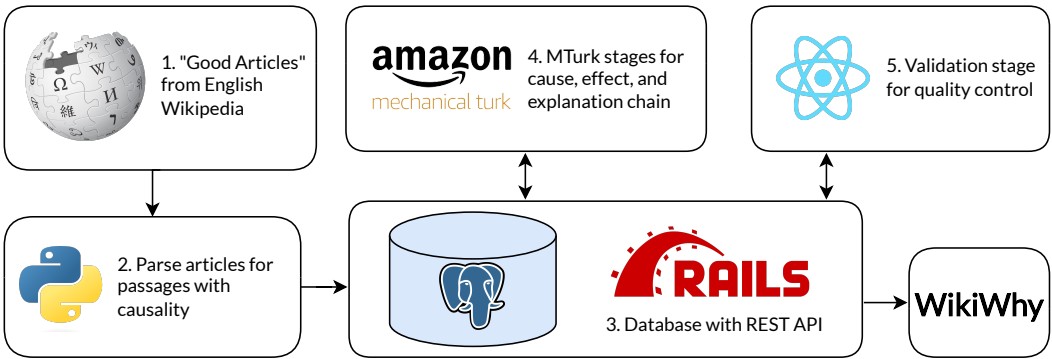

Figure 5: Dataset Collection and Validation Pipeline

**Cause**: There were time constraints to writing "Boruto: Naruto the Movie"
**Effect**: Hiroyuki Yamashita felt pressured writing "Boruto: Naruto the Movie"
**Explanation**: Creativity is difficult when put on a strict timetable. There was a need to both produce a good movie and do so on a strict time budget. These two demands put stress on Hiroyuki Yamashita while he worked.

**Cause**: Homer P. Rainey had liberal views.
**Effect**: Homer P. Rainey was fired by the University of Texas in 1944.
**Explanation**: If the University of Texas is conservative, they wouldn't want people working there who have liberal views.

**Cause**: the large size and reddish tint of red maple buds
**Effect**: Red maple buds which form in fall and winter are often visible from a distance.
**Explanation**: The color red stands out from a distance, so if the buds are red in the fall and winter, you'd be able to see them from a distance.

**Cause**: There were advances in technology, lower energy prices, a favorable exchange rate of the United States dollar, and lower alumina prices.
**Effect**: Productions costs of aluminum changed in the late 20th century.
**Explanation**: With advances in technology, prices of manufacturing change usually because they are now easier and cheaper to make. In this case it is aluminum that the price changed on because the technology improved the process.

Figure 6: GPT-3 Few-shot Exemplars

# Find a Cause-Effect Relation & Turn it into a Why Question.

**Your question must be specific enough to make sense ON ITS OWN (without the passage)**

| | |
|---|---|
| **Example 1: Use Full Names of People!** | ⌄ |
| **Example 2: Specify When and Where so the situation is totally clear!** | ⌄ |

## Your Passage (click link to find details)

https://en.wikipedia.org/wiki/Chrissie_Watts (link opens in new tab)

The aftermath dominated EastEnders in 2005 and helped to revive the fortunes of the show. According to the former head of BBC Drama Serials, Mal Young, this was dependent on the character of Chrissie, who was responsible for "anchoring the success of the anniversary storyline". A similar sentiment was expressed by Ian Hyland in the Sunday Mirror, who although critical of the convoluted plot felt EastEnders was improving "mainly `because` Chrissie is doing her best to rescue the fallout from the storyline dirty bomb Den's murder has become", and described the character as performing a "rescue act" on the show. However, Jim Shelley of the Daily Mirror was highly critical of Chrissie, calling her "the ludicrous Lady MacBeth wannabe", and felt her departure was ennabling EastEnders to move forward. In contrast, the TV editor of The Daily Telegraph Telegraph hailed Chrissie as "helping revive the show's fortunes that had been lagging somewhat in recent years".

click me only if the current passage lacks a cause-effect relation

## Step 1: Find Cause & Effect from the Passage

- To make the next step easier, write your effect as a full sentence with details so it's clear what the exact situation is
- Please do NOT repeat the cause in the effect box

**Cause:** write your specific cause here

**Effect:** write your specific effect here

## Step 2: Turn the Cause-Effect into a Why Question & its Answer

- **The Question should ask about Effect, and the Answer should be Cause**
  - Example: Cause=drug overdose, Effect=Heath Ledger died.
  - Question: Why did Heath Ledger die?
- **Your Question must make sense on its own (without the passage)**
  - Add details (who, what, when, where) so it's clear what the exact situation is
  - Use full names for people, groups, and places
- **Someone seeing only your QUESTION should NOT need to ask any clarifying questions**
  - They SHOULD NOT need to ask "which ___ are you talking about?"
    since your question should already be explicit about which ____ it's talking about

**Why Question:** write why question about effect here

**Answer (Cause):** [write cause & effect first!]

**Please Note**: Failing to follow instructions will result in your WorkerId being blocked from future tasks published by our group.
By submitting, you agree to the terms of this consent form.

Submit

Figure 7: Amazon Mechanical Turk Interface for Stage 1

## 1. Choose a Question to Answer (each involve cause & effect)

○ Why did "The similarity between The Fault in Our Stars and Perks of Being a Wallflower." lead to "Stephen Chbosky turned down the opportunity to direct The Fault in Our Stars."?

○ Why did "Stephen Chbosky turned down the opportunity to direct The Fault in Our Stars." result from "The similarity between The Fault in Our Stars and Perks of Being a Wallflower."?

○ Why does "The similarity between The Fault in Our Stars and Perks of Being a Wallflower." cause "Stephen Chbosky turned down the opportunity to direct The Fault in Our Stars."?

○ Why is "Stephen Chbosky turned down the opportunity to direct The Fault in Our Stars." a consequence of "The similarity between The Fault in Our Stars and Perks of Being a Wallflower."?

| Additional Context for the Questions | ⌃ |
|---|---|

On January 31, 2012, it was announced that Fox 2000, a division of 20th Century Fox, had optioned the rights to adapt John Green's novel The Fault in Our Stars for a feature film. Wyck Godfrey and Marty Bowen were due to produce the film with their production company, Temple Hill Entertainment. Stephen Chbosky, who directed The Perks of Being a Wallflower (also filmed in Pittsburgh), was in talks to direct the film but turned it down because of its similarity to Perks. On February 19, 2013, Josh Boone was hired as director; Scott Neustadter and Michael H. Weber were hired to adapt the novel into a screenplaytheir second adaptation for Fox, following Rosaline.

## 2. Answer the Question

| Example 1: Most explanations are fairly short (1 or 2 entries) | ⌄ |
|---|---|

| Example 2: Most explanations only require basic logic | ⌄ |
|---|---|

| Example 3: You may need to search online if you are unsure of the answer | ⌄ |
|---|---|

### Requirements
- Add a new entry for each step/sentence
- Use complete sentences with good spelling and grammar
- **Carefully read the question you chose and actually respond to it**
- **DO NOT** only write or rephrase "cause leads to effect"
  - We already know this! We want you to explain WHY that is the case

### Your Question
Why did "The similarity between The Fault in Our Stars and Perks of Being a Wallflower." lead to "Stephen Chbosky turned down the opportunity to direct The Fault in Our Stars."?

If he had directed both, it could have endangered his nomination for the Academy Award for "Perks of being a Wallflower"

Filming two similar movies would make him look like a one-trick pony.                    x

[ Add Explanation Step ]

**Please Note**: Submitting work with egregious grammar errors, inappropriately copied text, or nonsense answers (or otherwise failing to follow instructions) will result in your WorkerId being blocked from future tasks published by our group.
By submitting, you agree to the terms of this consent form.

[ Submit ]

Figure 8: Amazon Mechanical Turk Interface for Stage 2

