# OpenReview forum: "WikiWhy: Answering and Explaining Cause-and-Effect Questions"
_ICLR.cc/2023/Conference — ICLR 2023 notable top 5%_

### Official Review · Reviewer_V2He · 2022-10-24

**Confidence:** 3
**Correctness:** 3
**Technical Novelty And Significance:** 3
**Empirical Novelty And Significance:** 3
**Recommendation:** 8

**Clarity, Quality, Novelty And Reproducibility:**

* The paper is well-written and easy to follow. I particularly appreciate Figures 1 and 2, which give a good overview of how WikiWhy was constructed.
* There are many figures and tables in the appendices to help the reader understand the details of the paper and support reproduction. That being said, I cannot find all the hyper-parameters used for the GPT-2 Task 2 experiments.

**Strength And Weaknesses:**

## Strong

* The Cause and Effect QA with text generation problem is interesting (WikiWhy tasks 2 and 3)
* The automatic evaluation metric proposed is interesting.

## Weak

* (Minor) Table 1 has messy formatting. The Size and Topic columns should be right-aligned to support easy comparisons.
* Some hyper-parameters are missing (see the Clarity, Quality, Novelty, and Reproducibility section below).
* (Minor) The Conclusion states, "With this paper, we release WIKIWHY, a Question Answering dataset interested in understanding and facilitating the improvement of LLMs’ reasoning capability." The word "interested" is a strange choice for this sentence. I recommend rephrasing.

**Summary Of The Paper:**

The paper presents WikiWhy, a Question Answering dataset of "Why" questions constructed from Wikipedia text. WikiWhy contains questions requiring two steps of reasoning or based on a rationale set.

**Summary Of The Review:**

An interesting QA dataset. The paper has some minor issues.

---

> ### Author Response · Authors · 2022-11-11
> **Response to Reviewer V2He**
>
> We would like to thank the reviewer for the feedback. We have reformatted the table columns as suggested, and changed the phrasing in the conclusion to be more active “... a Question-Answering dataset **interested in understanding and facilitating**…” -> “a Question-Answering dataset **enabling the analysis and improvement of**...” Regarding fine-tuning GPT-2, we used standard hyperparameters for the Adam optimizer (learning rate = 1e-3, $\beta_1$ = 0.9, $\beta_2$ = 0.999, $\epsilon$ = 1e-8, decay = 0). These details have been added in the revision. Thank you again for your comments, and we are happy to address any other concerns.

---

### Official Review · Reviewer_reEb · 2022-10-25

**Confidence:** 3
**Correctness:** 3
**Technical Novelty And Significance:** 3
**Empirical Novelty And Significance:** 3
**Recommendation:** 6

**Clarity, Quality, Novelty And Reproducibility:**

Clarity: This work is generally well-written and easy to follow.

Quality: This work is technically sound.

Novelty: The novelty is not on the approach side but it is novel in terms of the task proposal, the dataset and the evaluation protocol.

Reproducibility: This work can be reproduced using the shared source code.



**Strength And Weaknesses:**

## Strength:
- It is the first work that investigates Q&A over cause-effect reasoning with explanations.
- The authors create a consolidated pipeline to define the problem and build the dataset, resulting a WIKIWHY dataset covering 11 topics.
-The authors propose new evaluation metrics for this cause-and-effect question answering task. Experiments verifies the reasoning ability of LLMs.

## Weakness:
- The problem formulation in which a sequence or a set of explanations are required make this task intrinsically difficult to evaluate.


**Summary Of The Paper:**

This paper proposes a new QA task on answering and explaining cause-and-effect questions. Different from previous approaches, the authors create the WIKIWHY dataset which contains the explanation within cause-effect relations. Taking the effect as the question source, the goal is to not only get the answer (cause) but also predict the explanations in sequential or set settings. Experiments show that the state of the art generative models are good at QA task but still a large space for improvement on the explanation task.



**Summary Of The Review:**

In general, this is an interesting work. Though there are some improvement space from the problem formulation and evaluation, this version of work is still good for the task proposal, dataset preparation and the experiment design.

---

> ### Author Response · Authors · 2022-11-13
> **Response to Reviewer reEb**
>
> Thank you for your time and feedback.
>
> Though our formulation does involve a challenging evaluation process, we believe that framing the explanation task as generative strengthens the benchmark as a proxy for simulated reasoning capability and also lays fertile ground for work on evaluation itself.
>
> While a retrieval version of the task would be easier to evaluate (and easier to perform) it would preclude the evaluation of few-shot and zero-shot reasoning abilities of decoder-only LMs like GPT-3 (objects of interest at this time in language-only reasoning) short of their performance on simple multiple-choice tests.
>
> Furthermore, building models for the reference-based generative language evaluation is an active and productive area of research. However, currently the majority of work in this area is focused on specific commercially-relevant tasks like translation and summarization (Rei et al. 2020; Xu et al. 2022). Generative, multi-step causal explanations is a novel generative language task that will enable future research along this direction into better sequence evaluation metrics that reflect human preferences such as quality, truthfulness, faithfulness, etc, along the lines we provide in our manual human evaluation.
>
> Our paper contributes a first step to solving this challenge of evaluating rationales with our proposed method that leverages encoder models to treat ideas as discrete units for ordered and unordered comparison. We have also considered a sentence-level ROUGE metric or NLI inspired evaluation (Xu et al. 2022). We look forward to experimenting with these methods and seeing how the community can build on this.
>
> Thank you again for your feedback!
>
> References:
> - Ricardo Rei, Craig Stewart, Ana C Farinha, and Alon Lavie. 2020. “COMET: A neural framework for MT evaluation” EMNLP 2020
> - Wenda Xu, Yilin Tuan, Yujie Lu, Michael Saxon, Lei Li, William Yang Wang. "Not All Errors are Equal: Learning Text Generation Metrics using Stratified Error Synthesis" EMNLP 2022

---

### Official Review · Reviewer_RHVP · 2022-10-26

**Confidence:** 4
**Correctness:** 3
**Technical Novelty And Significance:** 3
**Empirical Novelty And Significance:** 3
**Recommendation:** 8

**Clarity, Quality, Novelty And Reproducibility:**

As stated above, the paper presents the main ideas, the data collection procedure and the experimentation details with clarity.

The paper fills an important gap. Novelty or originality is not pertinent here with respect to the methodology.

There are adequate amount of details here and the main tasks are structured simply, so the data could be reproduced to some approximation with reasonable effort.

**Strength And Weaknesses:**

Strengths:

The paper introduces a high quality resource that fills a gap regarding evaluation of LLM’s ability to not only provide answers about causes and effects but also explain why using a wide variety of knowledge.

The paper is well-written and provides clear descriptions of how the data was collected, providing enough information to verify the quality of the resource.

The benchmarking experiments convincingly demonstrate the difficulty of the task. This resource could be a key driver for research into this problem space.


**Summary Of The Paper:**

Goal: The paper introduces a resource for explaining the cause-and-effect relationships on a wide variety of topics.

Method: The resource is crowdsourced in two steps. First, crowdworkers are asked to turn potential causal statements (identified via discourse cues) into Cause-effect questions. Then, the crowdworkers are asked to explain the cause effect relation to obtain “why” answers.

Using this process, a resource of about 9K why questions was curated from Wikipedia articles covering a wide variety of topics.

Benchmarking: The paper presents benchmarking results that show the

Key Contributions: The main contribution is in the resource itself. The resource presents a new test bed to assess abilities of NLP models to reason about cause and effect relations. The task requires a wide variety of knowledge and the benchmarking shows that the task is difficult for the current models and definitely warrants further research.


**Summary Of The Review:**

The paper makes an interesting contribution to explainable reasoning for cause-effect relations, which can drive interesting research into this space.

---

> ### Author Response · Authors · 2022-11-15
> **Response to Reviewer RHVP**
>
> Thank you for your notes. We found that previous reasoning tasks generally balance depth against evaluation difficulty and complexity against naturalness and scale. We aimed to address a gap and emphasize the integration of commonsense with a wide variety domain-specific facts through reasoning while keeping the aforementioned balance. We hope that this resource can foster further development in the space of reasoning and its evaluation, and that our transparent collection procedures can inform and aid the construction of future benchmarks.

---

### Official Review · Reviewer_tSWL · 2022-11-02

**Confidence:** 3
**Correctness:** 3
**Technical Novelty And Significance:** 2
**Empirical Novelty And Significance:** 3
**Recommendation:** 8

**Clarity, Quality, Novelty And Reproducibility:**

### Clarity:
The writing and presentation of the work are quite good. I had no issues understanding the basic ideas presented and experimental results. One issue is that there is some important content that has been pushed to the Appendix .

### Quality:
Overall, except for the protocol issue I outlined in question elicitation, I think the work is generally high quality,

### Novelty:
This work is moderately novel: explanations and reasoning is not a new topic in QA (see the related work along with my suggested references). Nonetheless, the dataset resource, along with proposed task formulation should be a worthwhile addition to this area.

### Reproducibility:
As far as I can tell, most of the resources for annotation and baselines experiments are made available. The one lacking resource is in Section 5.3 (Human Evaluation) which used student raters. The specific interface was not made available (like the MTurk one from Appendix A4), but perhaps it was just a spreadsheet?


**Details Of Ethics Concerns:**

The work described in this paper involves the construction of datasets that involved both MTurk workers and student raters. While the paper contains an 'Ethics Statement' describing some of the aspects of the work, I am not knowledgeable to determine if the information is sufficient.

**Strength And Weaknesses:**

## Strengths

1. The paper addresses an important weakness in many existing question-answering datasets that focus on answering questions with limited feedback on system reasoning or explanations. The proposed formulation of explicitly requiring systems to produce causal explanations to “why” questions is an interesting step,

2. The proposed task formulation is intuitive and elegant. While the two types of explanations proposed (step sequence and sets) are likely not comprehensive, they probably cover a large proportion of explanations to simple ‘why’ questions,

3. Collecting datasets is challenging, but can be helpful directing improvements to future systems. The release of this dataset to the community will likely help drive new work in QA explanations and reasoning.

4. The baselines described in the paper are quite reasonable starting points for modeling,

5. The human judgements of the baselines are surprisingly low which means that this task has large room for improvement in future work. (Note to authors: it would be nice if the baseline predictions were released along with your code+dataset so that they can be inspected by the community).

## Weaknesses

1. The main weakness in this work relates to the protocol used to elicit ‘why’ questions conditioned on a passage that contains a causal connective. Unfortunately, I believe this contains some of the same issues the community has observed with SQuAD task, namely: (1) generated questions are artificial since conditioned a specific passage, and rarely reflect naturally occurring ‘why’ questions, and (2) unusually high overlap in phrasing between the question and the answer passage, and (3) reliance of passage context for question to be fully interpretable. Fortunately, the paper addresses issue (3), but still has issues (1) and (2).

   Looking over most questions in the dataset, they mostly contain the issues listed. Examples:

   a) Question: "Why did the member of The Cure, Lol Tolhurst, make minimal contributions to the album Disintegration?"

     Passage excerpt: “... revealed that while Tolhurst had contributed to the song "Homesick", his contributions to the rest of the album were minimal due to his alcoholism”


    b) Question: “"Why is it unlikely that any sequels would be made for the Power Rangers film?"

      Passage excerpt: “However, in May 2017, Forbes noted that due to the underwhelming performance of the film in most markets, it was unlikely any sequels would be made.”

    In these examples (and others I browsed) it seems that (1) questions are formulated slightly unnaturally, (2) very high word overlap between answer phrase and question, (3) questions are all answerable within a single sentence.

    It is probably too late and costly to ask authors to revise their protocol for question generation. For future reference, one protocol that avoids the pitfalls of SQuAD is the TyDiQA dataset (https://arxiv.org/pdf/2003.05002.pdf).

 2. Probably due to the issue above, the retrieval performance using BM25 is unusually high  (Section 5.1). For comparison, the “Deep Passage Retrieval” paper (https://arxiv.org/pdf/2004.04906.pdf) shows BM25 performance in the 40s for Natural Questions. Similarly, answer identification should be quite easy as well for a model fine-tuned on this data given the high word overlap between answer phrase and question.

   So while this dataset will be useful to understand whether models are able to produce good reasoning/explanations for certain types of ‘why’ questions, it is, unfortunately, not a good dataset for QA retrieval and answer identification.

3. There is a part of the human evaluation (Section 5.3), specifically the Win/Tie/Lose which seems problematic. In our previous experience, disclosing to raters which outputs are “reference” and which are “system” biases ratings measurably. It is typically needed to make this type of analysis blind by (1) randomly shuffling which output is reference vs. system when presenting them to users, and (2) never disclosing the source of the explanation (reference or system).

## Other notes / suggested citations or references:

1. A dataset that seems quite related is ELI5 (https://arxiv.org/abs/1907.09190 , https://facebookresearch.github.io/ELI5/) which also aims to measure model’s ability to generate QA explanations.

2. Another dataset is the TellMeWhy (https://arxiv.org/pdf/2106.06132.pdf) which asks models to produce simple, reasoning-based explanations for answers in the context of stories.

3. One other dataset that I thought worth mentioning in the related work is e-SNLI (https://arxiv.org/pdf/1812.01193.pdf , https://github.com/OanaMariaCamburu/e-SNLI ). While not strictly a QA task, the explanations collected in e-SNLI are quite related to the ones presented in this work.

4. Another system that should be referenced is WT5 (https://arxiv.org/pdf/2004.14546.pdf) which is a T5-based system for generating explanations/rationales. Some of the checkpoints are available at https://github.com/google-research/google-research/tree/master/wt5#released-model-checkpoints and could serve as a stronger baseline than GPT-2, if fine-tuned for your dataset/task.


**Summary Of The Paper:**

## Summary

This paper introduces a new dataset, WikiWhy, along with an explanation task, that aims to measure the reasoning and common sense abilities of generative language models. The work describes the motivation for this new dataset and task, mainly that most existing question-answering tasks require limited reasoning and common-sense knowledge. The paper describes how the new dataset was collected, based on Wikipedia with a couple of MTurk tasks: (1) generating question-answer pairs given a specially selected passage, and (2) generating a series of sentence-oriented rationales. The paper describes experiments using baseline models (based on GPT-2 with fine-tuning, and GPT-3 with prompting), demonstrating low performance on the proposed task. Finally, the paper proposes an automatic metric for evaluating system outputs on the new task, demonstrating correlation of the metric with human judgements.

**Summary Of The Review:**

Overall I’m leaning to recommend acceptance of this paper. It has become clear to the QA community that we should move beyond factoid seeking queries and “exact match” as an evaluation metric. This paper proposes an interesting task variant. Despite some of the flaws in the data collection protocol (see above), I think this resource will be of interest to the community. The task formulation may also be extended in the future for other types of rationales. Finally, the initial baseline experiments indicate a large gap for improvement.

---

> ### Author Response · Authors · 2022-11-15
> **Response to Reviewer tSWL**
>
> We’d like to thank the reviewer for detailed notes and feedback.
> ### Data Collection Methodology
>
> Our initial approach involved asking annotators to freely come up with “why” questions. While this guaranteed naturalness to the why questions, we ran into issues of question diversity and a lack of in-article/on-Wikipedia answer grounding (annotators often answered with their own knowledge and cited articles at random). Without this grounding enabling second-stage annotators to provide quality explanations was infeasible. After all, when the vague notion of a “why” question is elicited to an annotator without further guidance, many cliche topics like “why is the sky blue” are extremely prevalent. We believe this general lack of diversity in randomly elicited “why” questions is connected to the general lack of “why” questions in open domain QA datasets in the first place. When we adapted to a TyDiQA-like open annotator-curiosity-driven QA pair collection methodology, we found that these issues continued to persist. The simple fact is there are neither many “why” questions circulating in annotator’s heads nor readily available in a random sample of Wikipedia text, without performing considerable filtering beforehand. Unfortunately, contrived QA pairs are inevitable for this task.
>
> However, though we agree with you that our QA pairs do not reflect the distribution of user-relevant questions, we believe that their contrived nature (which is a consequence of their text-grounding) is what makes them particularly well-suited for assessing reasoning capabilities. By motivating our dataset’s explanations in real cause->effect relationships which can be posed as open-domain “why” questions from known publicly-available text on Wikipedia, we produce a benchmark for natural language reasoning that is very fair to generative models in a zero-shot environment—the necessary information for explaining the relationship was present at training, but is also challenging in its open-domain nature. This is a contrast to the contrived, limited-domain reasoning benchmarks from prior work that are limited to math or science test questions.
>
> To summarize, while we agree that the high Q-A phrasing overlap and reliance on external passage context for interpretability of the proposed answers limit our resource’s value as a standalone QA benchmark, these casualties are what give rise to its novelty as a resource for benchmarking LM reasoning. Treating the QA pairs as stimuli for producing quality {cause, effect, explanation} triples is our core contribution.
>
> ### Evaluation Updates
>
> For the experiment sections, we appreciate your suggestion on the win-tie-lose human evaluation. To clarify, the human evaluations were conducted via spreadsheets. In our original evaluation setup, we did not disclose which column was the gold reference and which was the generated candidate, but we did not shuffle the two. Following your suggestion, we have re-done this experiment with the shuffling and the updated paper revision contains the results (Table 3).
>
> Before (Left), After (With Shuffling, Right)
> | Method       |  Win  |  Tie  |  Lose  | \\  |  Win  |  Tie  |  Lose  |
> |--------------|-------|-------|--------|-----|-------|-------|--------|
> | GPT-2 (EO)   | 0.000 | 0.060 | 0.940  | \\  | 0.040 | 0.040 | 0.920  |
> | GPT-3 (EO)   | 0.220 | 0.200 | 0.580  | \\  | 0.080 | 0.360 | 0.580  |
> | GPT-3 (A&E)  | 0.020 | 0.120 | 0.860  | \\  | 0.080 | 0.100 | 0.820  |
>
>
>
> ### Other Additions
>
> Thank you for the additional references, we’ve included discussion and citations in the most recent revision. With respect to WT5, we could not run WT5-11B. We requested the weights for WT5-Base from the authors, but we have not heard back in time. We are attempting to complete a replication of WT5-Base and include results in the camera-ready version. Results or not, we will cite WT5 as an important related work. Per your suggestion, we've included the baseline model predictions with our supplementary materials.
>
> ### In Summary
>
> Thanks again for your input. We truly appreciate your time and suggestions!

---

### Author Response · Authors · 2022-11-19
**General Response**

We want to thank all reviewers for their insightful feedback. The revised paper for this stage has been uploaded. We have included the following: additional discussion and references to related work, hyperparameters for tuning GPT-2, updated results for win-tie-lose human evaluation, and light formatting changes. For more detailed elaboration on these points of feedback, please refer to the comment replies under the relevant review. We’re looking forward to hearing your comments. In summary, we agree with your assessments that there is a gap in resources for evaluating explicit reasoning *between* cause-effect relations and that WikiWhy can fill this gap and drive future progress in the directions of reasoning and interpretability. We hope to foster a better understanding of the models’ capabilities and inspire work in generating and evaluating explanations. Thank you again for your feedback and consideration.

---

> ### Comment · Area_Chair_jRyH · 2022-11-28
> **Please look at the updated draft**
>
> Dear reviewers,
>
> Could you please take a look at the updated draft uploaded by the authors and see if you want to change your scores?
>
> Thanks!

---

### Decision · Program_Chairs · 2023-01-20

**Decision:**

Accept: notable-top-5%

**Justification For Why Not Higher Score:**

n/a

**Justification For Why Not Lower Score:**

There is no particular reason to reject the paper or provide a lower score as shown by the strong agreement among the reviewers. Further, the authors have addressed most of the concerns in the updated version of the paper.

**Metareview: Summary, Strengths And Weaknesses:**

This paper presents a new dataset of question answering that also focuses on selecting the reasons for a particular answer. In particular, the authors observed a lack of "why" questions in existing datasets, and created a dataset that focuses on these questions, while also adding a concept of explaining why a particular answer was chosen for the question. The reasoning is required to be in the form of a few sentences that when put together explain the answer. The authors also showed that existing LLMs do not work on this task out of the box hence demonstrating the difficulty of the task.

Strength
1) Very clearly written paper with thorough explanation of data collection methodology
2) Release of a dataset that is useful for the community and will advance research in reasoning
3) Establishment of baseline experiments for QA and QA+reasoning on this dataset

Weakness
1) There is some issue with the nature of the "why" questions because these are not totally open-ended but rather very focused on the text present in the answer passages. However, this cannot be easily rectified as soliciting open-ended questions from annotators come with its own set of challenges.

**Note From Pc:**

if the above contains the word "oral" or "spotlight" please see: "oral" presentation means -> notable-top-5% and "spotlight" means -> notable-top-25%. As stated in our emails, we are disassociating presentation type from AC recommendations

**Summary Of Ac-Reviewer Meeting:**

n/a